# Long-term outcomes of prismatic correction in partially accommodative esotropia

**Hye Rim Choe** [ʘ], **Hee Kyung Yang** [ʘ], **Jeong-Min Hwang** [ʘ] *

Department of Ophthalmology, Seoul National University College of Medicine, Seoul National University Bundang Hospital, Seongnam, Korea

ʘ These authors contributed equally to this work.
* hjm@snu.ac.kr

**Data Availability Statement:** All relevant data are within the manuscript and its Supporting Information files.

**Funding:** The authors have no funding or sources of support received during this specific study.

## Abstract

### Purpose

In partially accommodative esotropia (PAET), prism glasses can correct small angles of residual esotropia but the long-term effect of prismatic correction alone without surgery has not been reported. We aimed to investigate the long-term outcome of prism glasses after full hypermetropic correction for PAET.

### Methods

This retrospective, case-control study was performed for children aged 10 years or younger with a residual esotropia of ≤ 20 prism diopters (PD) after full hypermetropic correction who were fitted with prism glasses and followed-up for 3 years or more. Clinical characteristics and the angle of esodeviation were obtained at each follow-up examination. Successful motor outcome after 3 years of prismatic correction was determined if the residual angle of esotropia after full hypermetropic correction was ≤ 10PD. Patients who eventually weaned off prism glasses were noted.

### Results

Among 124 patients, 30.6% achieved success and 7.3% weaned off prism glasses after 3 years of prism-wear. Smaller amount of latent esodeviation (P = 0.001) revealed by prism adaptation and good fusional response at near with the Worth 4-dot test were significant prognostic factors of success by multivariate analysis (P = 0.033). After 3 years of wearing prism glasses, the rate of improvement in stereoacuity was higher in the Success group (60.5% vs 27.9%) (P = 0.001), however, there was no significant difference between the prism-weaned group and prism-wearing group within the Success group (P>0.05).

### Conclusion

Prism glasses for small angle PAET can be a treatment option in patients who have a small angle of latent esodeviation revealed by prism adaptation and good sensory fusion at near. Otherwise, early surgery may be advisable as the majority of patients showed suboptimal outcome even after long-term prism-wear.

**Competing interests:** The authors have declared that no competing interests exist.

## Introduction

Partially accommodative esotropia (PAET) is a form of accommodative esotropia characterized by residual esodeviation after full correction of hypermetropia.[1–3] The first-line treatment of PAET is full correction of hypermetropic refractive errors, then various therapies, including occlusion, prism, miotics, botulinum toxin injection or surgical intervention, are considered to treat the remnant deviation.[4–6]

In patients with PAET, surgery is usually considered if fusion cannot be achieved after 6–8 weeks of hypermetropic correction or if the remnant deviation is greater than 10 prism diopters (PD) both at near and at distance with full correction of hypermetropia.[6] Occasionally, prisms have been used in the management of PAET. Prism adaptation using prism glasses may determine the amount of surgery needed, as without it, it frequently results in under-correction or postoperative surgical drift with a sudden increase of esotropia after surgery.[7–13] In a previous study of acquired esotropia, alignment was stabilized and sensory fusion was achieved when prism glasses were administered, and the frequency of postsurgical drift decreased, showing a good prognosis.[10] In the 1-year follow-up study investigating patients with PAET, orthotropia or esophoria was maintained with prism glasses without surgery in 44% of patients, and none of them showed deterioration of stereoacuity.[14] Additionally, in consecutive esotropia after exotropia surgery, prismatic correction achieved a successful motor alignment with good stereoacuity.[15, 16]

Based on these previous studies, it can be expected that prismatic correction of esodeviation might be helpful for maintaining fusion and improving binocular sensory status while hopefully, sufficient exodrift occurs to reduce the amount of esodeviation. However, to the best of our knowledge, the long-term results of the use of prism glasses in PAET have not been reported. Therefore, in this study, we aimed to investigate whether long-term use of prism glasses alone can reduce the amount of esodeviation and improve sensory status in PAET with residual esotropia after full hypermetropic correction.[15, 16]

## Materials and methods

A retrospective review was performed on children aged 10 years or younger with a residual esotropia of $\leq 20$ PD at distance measured by the simultaneous prism and cover test (SPCT) after full hypermetropic correction between June 2003 and December 2012.

Patients who were fitted with prism glasses to correct their residual esodeviation and followed-up for 3 years or more were recruited. Patients with hypermetropia less than +1.50 diopters, history of prior strabismus surgery, dissociated vertical deviation, oblique muscle dysfunction, paralytic or restrictive esotropia, ocular pathology, chromosomal anomaly, neurological disorder or developmental abnormality were also excluded. All patient records were de-identified and analyzed anonymously.

We investigated age, gender, best corrected visual acuities (BCVA), refractive errors with cycloplegic refraction, angle of esotropia, and binocular sensory status. All patients underwent full ophthalmologic examination. Cycloplegic refraction was performed after the administration of 1% cyclopentolate. The angle of esodeviation was measured by the prism alternate cover test at 6 m and 0.33 m. When an observable difference was noted on the cover test, SPCT was performed to measure the angle of manifest tropia.[17] Measurements of ocular deviation were performed by one examiner (J-MH). Prism glasses were initially prescribed fully correcting the cycloplegic refractive errors and residual angle of esotropia during distant fixation (DCC$_{Initial}$). The amount of prisms was changed whenever the angle of residual esotropia was changed. For example, if a patient's cycloplegic refractive errors were +3.00 Dsph in both eyes, and the esodeviation measured with hyperopic glasses (+3.00 Dsph OU) was 12 PD during distant fixation,

then glasses of +3.00 Dsph with 6 PD base-out prisms were prescribed for each eye. At the next follow-up, if the patient revealed an esotropia of 4 PD wearing those glasses with the SPCT, the amount of prisms were increased to 8 PD base-out prisms for each eye. Initial prism adaptation was performed up to 3 months until the esodeviation angle stabilized, and the maximum angle of esodeviation was noted as the baseline angle of esodeviation ($DCC_{PAT}$, $NCC_{PAT}$) which represents the total amount of latent strabismus. The amount of total esodeviation at each follow-up examination was calculated as the summation of the measured angle of esotropia wearing prism glasses and the amount of base-out prisms on both eyes.

Accommodative-convergence over accommodation (AC/A) ratio was measured at the initial examination by the clinical method evaluating distance-near relationship. High AC/A ratio was defined when the near esodeviation was $\geq 10PD$ larger than the distant esodeviation. Amblyopia was defined as a difference in BCVA between both eyes of more than two lines, and for those with this, occlusion therapy for the fellow eye was recommended. Anisometropia was defined as a difference in cycloplegic refractive errors of 2 diopters or more in spherical equivalent values between fellow eyes. Binocular sensory status was examined with full hypermetropic correction. Randot stereoacuity test (StereoOptical Co., Inc. Chicago, IL) and Worth 4-dot (W4D) test were tested at 6 m and 0.33 m in children older than 30 months of age. Improvement of stereopsis with the Randot stereoacuity test was defined as a change of 2 octaves or more.[18] The results of the W4D test were described as either normal (the patient sees all four dots) or abnormal, including diplopia (the patient sees five dots) and suppression (the patients sees 3 green dots or 2 red dots). The W4D test results measured during the first 3 months of prism adaptation were noted as the baseline results.

Patients were examined at 1, 3, 6, 12, 24 and 36 months after wearing prism glasses. Glasses were adjusted according to their refractive errors or residual esotropia after full hypermetropic correction. Successful motor outcome was determined if the amount of residual esotropia after full hypermetropic correction (the summation of the measured angle of esotropia wearing prism glasses and the amount of base-out prisms on both eyes) was $\leq 10$ PD, and fusion was maintained with prism glasses at the final examination. Otherwise, subjects were classified into the Failure group, including those who underwent strabismus surgery before 3 years of prismatic correction. Surgery was performed in patients who showed an esotropia of $\geq 10PD$ even after wearing prism glasses prescribed at the previous examination. The decision for surgery was made after wearing prism glasses at least one year. Patients who eventually weaned off prism glasses as they showed orthotropia or only phoria without prism glasses during follow-up were also noted.

To compare the characteristics between the two groups for categorical variables, Fisher's exact test or chi square test were used. For continuous variables, Mann-Whitney U test or independent student t-test were used. When comparing the stereoacuity over time in both groups, age was treated as a covariance, and the analysis of covariance (ANCOVA) test was used. Prognostic factors associated with success after wearing prism glasses were estimated using univariate and multivariate logistic regression analyses. Statistical analyses were performed using SPSS software for Windows version 22.0 (SPSS, Inc., Chicago, IL). A *P*-value of less than 0.05 was considered statistically significant. This study adhered to the tenets of the Declaration of Helsinki. Institutional ethics review board approval (B-1604-343-101) was obtained for this study by Seoul National University Bundang Hospital.

## Results

A total of 124 PAET patients were included. During the 3 years of follow-up, 86 patients (69.4%) were classified as the Failure group and 38 patients (30.6%) as the Success group. Of

the 86 patients in the Failure group, 21 patients (24.4%) received surgery, while the other patients who did not meet the surgical criteria maintained their glasses with prisms. The mean duration from the initial examination performed at first visit to the time of operation was 1.8 ±0.7 years (range, 1.0~2.8).

At the initial examination, amblyopia was present in 27 patients (21.8%). Full-time occlusion except one day in a week was prescribed in 21 patients (16.9%), while six patients (4.8%) who were less than 4 years old performed part-time occlusion therapy for 2 to 6 hours a day. Occlusion was tapered according to their visual response, and 8 of patients (6.5%) continued part-time occlusion until the last follow-up examination. A total of 17 patients (13.7%) showed a high AC/A ratio at the initial examination.

Of the 38 patients in the Success group, prism glasses were weaned off in 9 patients (23.7%) and these patients finally showed orthotropia or esophoria without prismatic correction. After 3 years of follow-up, the final angle of esodeviation was 5.6±3.8 PD at distance and 5.9±4.4 PD at near in the Success group.

Table 1 compares the clinical characteristics between the two groups. There were no significant differences between the groups with respect to gender, age of initial visit, BCVA, and refractive errors. The residual angle of esodeviation at distance and near with full hypermetric correction ($DCC_{Initial}$, $NCC_{Initial}$) showed no significant differences. The maximum angle of baseline esodeviation with full hypermetropic correction ($DCC_{PAT}$, $NCC_{PAT}$) was determined by prism adaptation after stabilization of the esodeviation with 3 months of prism-wear. The maximum angle of baseline esodeviation revealed by prism adaptation ($DCC_{PAT}$, $NCC_{PAT}$) were greater in the Failure group than in the Success group ($P = 0.011$ at distance and $P < 0.001$ at near).

Regarding binocular sensory status, the W4D test results at near were significantly better in the Success group at baseline ($P = 0.012$) and after 3 years ($P = 0.008$) showing a higher rate of binocular fusion. On the other hand, the W4D results at distance did not show significant difference between the Success group and Failure group at baseline ($P = 0.092$) and final examinations ($P = 0.559$). Among the patients who showed an abnormal response with the W4D test at baseline, 37.3% of patients showed a fusional response at distance and 32.3% at near after 3 years. The rate of improvement with the W4D test were not significantly different between the two groups at distance (45.5% in Success group and 60.9% in Failure group) ($P = 0.316$) and at near (55.6% in Success group and 42.9% in Failure group) ($P = 0.378$).

Regarding the prism-adapted motor response, the majority of patients required an increase in prism dosage. The maximum change in esodeviation after 3 months of prism adaptation was significantly larger in the Failure group compared with the Success group at distance (+7.9 ±5.1 PD vs +5.2±4.8 PD, $P = 0.008$) and at near (+5.2±5.9PD vs +2.0±4.6 PD, $P = 0.009$).

Univariate and multivariate analyses of prognostic factors associated with motor success after prism-wear are shown in Table 1. Occlusion therapy, presence of amblyopia, Randot stereoacuity results, and the baseline W4D test results at distance did not significantly affect motor success according to univariate analyses.

By multivariate logistic analysis, a good fusional response with the W4D test at near ($P = 0.043$, Odds ratio = 3.049, 95% confidence interval 1.03–9.35) and a smaller amount of change in the maximum angle of esodeviation at distance revealed by baseline prism adaptation ($\Delta DCC_{PAT}$) ($P = 0.005$, Odds ratio = 0.849, 95% confidence interval 0.75–0.95) were significant predictive factors of motor success after prismatic correction.

After 6 months of wearing prism glasses, the angle of esodeviation gradually decreased more in the Success group than in the Failure group for up to 3 years both at distance and at near (Fig 1). The annual changes in esodeviation at distance were -3.8±1.9 PD/year in the Success group and +0.7±3.7 PD/year in the Failure group ($P < 0.001$). The annual changes in

**Table 1. Clinical characteristics of patients according to the final motor outcome of prismatic correction.**

| | Failure group | Success group | *P*-value | Multivariable logistic analysis | | |
|---|---|---|---|---|---|---|
| | | | | OR | 95% CI | *P*-value |
| Number of patients | 86 (69.4%) | 38 (30.6%) | | | | |
| Sex (male) | 48 (55.8%) | 16 (42.1%) | 0.112[a] | | | |
| Age of initial visit (years) | 4.8±2.6 (0.9~10.0) | 4.7±2.2 (1.3~10.0) | 0.843[b] | | | |
| Age of onset (years) | 3.8±2.3 (0.1~9.8) | 3.6±2.1 (1.0~10) | 0.791[b] | | | |
| BCVA (logMAR) | 0.30±0.27 (-0.08~0.82) | 0.31±0.28 (-0.08~0.82) | 0.589[b] | | | |
| SEQ (D) | +4.17±1.65 (+2.00~+9.00) | +4.26±1.37 (+1.50~+7.00) | 0.089[c] | | | |
| Stereoacuity (log arcsec)* | 2.40±0.65 (1.78~3.54) | 2.95±0.79 (1.85~3.54) | 0.070[d] | | | |
| $DCC_{Initial}$ (PD) | 12.7±3.3 (4~20) | 12.3±3.5 (4~20) | 0.456[b] | | | |
| $NCC_{Initial}$ (PD) | 14.6±5.7 (4~33) | 12.7±6.1 (1~26) | 0.109[b] | | | |
| $DCC_{PAT}$ (PD) | 20.0±5.8 (10~34) | 17.2±5.1 (10~28) | **0.011**[b] | | | |
| $NCC_{PAT}$ (PD) | 19.2±6.9 (0~34) | 14.5±6.1 (1~28) | **<0.001**[b] | | | |
| $\Delta DCC_{PAT}$ (PD) | +8.0±5.1 (0~18) | +5.2±4.8 (0~16) | **0.008**[b] | 0.849 | 0.75–0.95 | **0.005** |
| $\Delta NCC_{PAT}$ (PD) | +5.3±5.9 (0~23) | +2.1±4.6 (0~20) | **0.009**[b] | | | |
| $W4D_{PAT}$ normal at distance[†] | 35 (46.7%) | 19 (63.3%) | 0.092[e] | | | |
| $W4D_{PAT}$ normal at near[‡] | 31 (37.3%) | 18 (64.3%) | **0.012**[e] | | | |
| Occlusion therapy | 69 (80.2%) | 28 (73.7%) | 0.278[e] | | | |
| Amblyopia | 36 (41.9%) | 14 (36.8%) | 0.374e | | | |
| Anisometropia | 8 (9.3%) | 0 (0%) | **0.048**[e] | | | |
| $DCC_{Final}$ (PD) [f§] | 20.2±4.7 (12~30) | 5.6±3.8 (-4~10) | **<0.001**[b] | | | |
| $NCC_{Final}$ (PD) [f§] | 15.1±8.2 (0~40) | 5.9±4.4 (-4~18) | **<0.001**[b] | | | |
| $W4D_{Final}$ normal at distance [f¶] | 45 (80.4%) | 27 (79.4%) | 0.559[e] | | | |
| $W4D_{Final}$ normal at near [f#] | 36 (58.1%) | 30 (83.3%) | **0.008**[e] | 3.049 | 1.03–9.35 | **0.043** |

*P*-values marked in bold indicate numbers that are significant on the 95% confidence limit; BCVA = best-corrected visual acuity; SEQ = spherical equivalent of cycloplegic refractive error; D = diopters; PD = prism diopters; PAT = prism adaptation test; $DCC_{Initial}$ = distant esodeviation with full hypermetropic correction before prism-wear; $NCC_{Initial}$ = near esodeviation with full hypermetropic correction before prism-wear; $DCC_{PAT}$ = the maximum baseline angle of esodeviation at distance during the first 3 months of prism-wear with full hypermetropic correction; $NCC_{PAT}$ = the maximum baseline angle of esodeviation at near during the first 3 months of prism-wear with full hypermetropic correction; $\Delta DCC_{PAT}$ = $DCC_{PAT}$-$DCC_{Initial}$;$\Delta NCC_{PAT}$ = $NCC_{PAT}$-$NCC_{Initial}$;$W4D_{PAT}$ = Worth-4-dot test results during the first 3 months of prism-wear with full hypermetropic correction; $DCC_{Final}$ = distant esodeviation with full hypermetropic correction (the summation of the measured angle of esotropia wearing prism glasses and the amount of base-out prisms on both eyes) after 3 years of prism-wear; $NCC_{Final}$ = near esodeviation with full hypermetropic correction (the summation of the measured angle of esotropia wearing prism glasses and the amount of base-out prisms on both eyes) after 3 years of prism-wear; $W4D_{Final}$ = Worth-4-dot test results with full hypermetropic correction after 3 years of prism-wear;

[a] Chi square test;

[b] Mann-Whitney U test;

[c] Independent student t-test;

[d] Analysis of covariance (ANCOVA) test;

[e] Chi square test;

[f] In the failure group, 21 patients who underwent strabismus surgery before 3 years of follow-up were excluded;

*N = 45 (32/86 of the Failure group and 13/38 of the Success group were initially tested due to young age);

[†]N = 105 (75/86 of the Failure group and 30/38 of the Success group were tested);

[‡]N = 111 (83/86 of the Failure group and 28/38 of the Success group were tested);

[§]N = 103 (65/65 of the Failure group and 38/38 of the Success group were tested);

[¶]N = 90 (56/65 of the Failure group and 34/38 of the Success group were tested);

[#]N = 98 (62/65 of the Failure group and 36/38 of the Success group were tested)

**Distance deviations with hypermetropic correction**

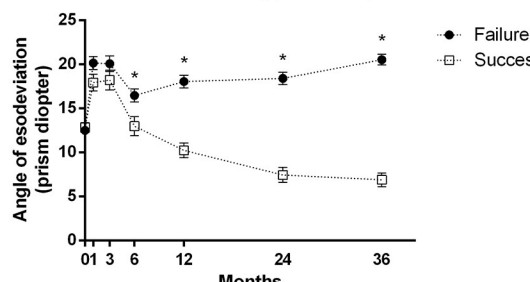

**Near deviations with hypermetropic correction**

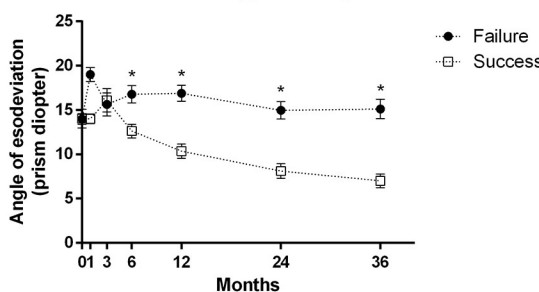

**Fig 1. Change in esodeviation at distance and near according to the duration of prism-wear.** The angle of esodeviation was calculated after full hypermetropic correction. After prism-wear, the summation of the measured angle of esotropia with prism glasses and the total amount of base-out prisms were noted. (Left) During distance fixation, the angle of esodeviation at distance became smaller in the Success group compared to the Failure group after 6 months ($P$ = 0.019), and this difference was maintained up to 3 years. Annual changes of esodeviation during distance fixation compared to baseline were -3.8±1.9 PD/year in the Success group and +0.7±3.9 PD/year in the Failure group ($P$<0.001). (Right) During near fixation, the angle of esodeviation at near became smaller in the Success group compared to the Failure group after 6 months ($P$ = 0.009), and this difference was maintained up to 3 years. Annual changes of esodeviation during near fixation were -2.9±2.1 PD/year in the Success group and +0.2±5.2 PD/year in the Failure group ($P$<0.001). (*$P$<0.05).

esodeviation at near were -2.9±2.1 PD/year in the Success group and +0.2±5.2 PD/year in the Failure group ($P$ < 0.001). In the Failure group, the annual change in esodeviation of those who received surgery (+3.1±5.7 PD/year at distance and +4.9±7.3 PD/year at near) was larger than those who did not (-0.1±2.3 PD/year at distance and -1.3±3.1 PD/year at near) ($P$ < 0.001 at distance and at near).

Randot stereoacuity did not show a significant difference between both groups at the initial examination. After 3 years, stereoacuity was significantly better in the Success group than in the Failure group ($P$ = 0.010) (Fig 2). After 3 years of wearing prism glasses, or at the time of surgery, 37.9% of patients showed improvement in stereopsis by more than 2 octaves. The rate of significant improvement in stereopsis by more than 2 octaves was significantly higher in the Success group (60.5%) than that in the Failure group (27.9%) ($P$ = 0.001).

Among the 38 patients in the Success group, 9 patients (23.7%) finally weaned off prism glasses within 3 years, and the mean duration of wearing prism glasses was 28±10.3 months (range, 12~36). Subgroup analysis was performed to compare the characteristics between patients who weaned off prism glasses within 3 years or not in the Success group (Table 2). The initial angle of esodeviation after full hypermetropic correction at near before wearing prism glasses (NCC$_{Initial}$) as well as the near angle of esodeviation after prism adaptation (NCC$_{PAT}$), was significantly smaller in the prism-weaned group which did not exceed 15 PD (Table 2). Meanwhile, Randot stereoacuity did not show significant difference between the prism-weaned group and prism-wearing group on the initial ($P$ = 0.147) and final examination after 3 years ($P$ = 0.445). The rate of improvement in stereoacuity was not statistically different between the two groups (62.1% in Prism-wearing group vs 55.6% in Prism-weaned group) ($P$ = 0.510). The flow chart of results according to the response to prismatic correction is presented in Fig 3.

## Discussion

In this study, we showed the long-term effect of prism glasses for patients with PAET when the amount of residual esotropia is 20PD or less after full hypermetropic correction. Approximately 30.6% of patients did not require surgery and finally showed a residual esotropia (the summation of the measured angle of esotropia wearing prism glasses and the amount of base-

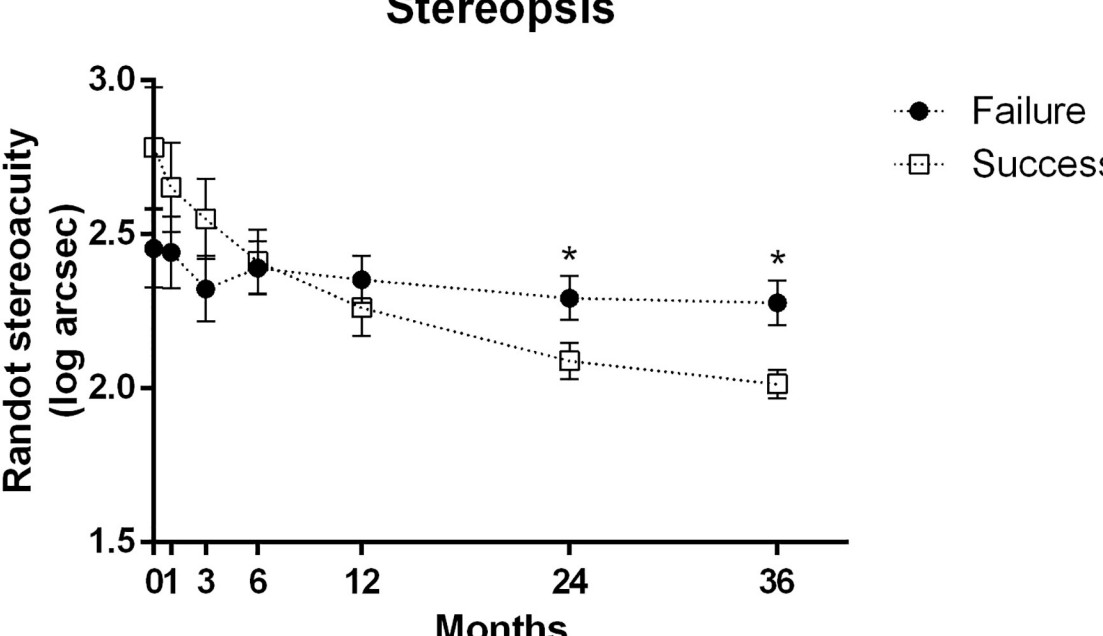

**Fig 2. Change in stereopsis according to the duration of prism-wear.** The difference in stereoacuity between the Success group and Failure group gradually increased. Stereoacuity was significantly better in the Success group than the Failure group after 3 years ($P = 0.010$) of prism-wear. ($^*P<0.05$).

out prisms on both eyes) of $\leq 10$ PD after 3 years. Smaller amount of latent esodeviation revealed by prism adaptation, and a good fusional response at near were strong prognostic factors of success. Finally, 7.3% of patients weaned off prism glasses within 3 years of prismatic correction.

The first-line treatment for PAET is full hypermetropic correction; surgery is commonly performed for the treatment of residual esodeviation.[4–10] Occasionally, prism glasses are recommended for small angles of esodeviation.[4, 5] The purpose of this study was to investigate whether the correction of residual esotropia and fusional response can be achieved by incorporating the use of prism glasses with full hypermetropic correction to improve the maintenance of binocular alignment and reduce the angle of esodeviation, with hopes of this method eventually replacing surgery. Furthermore, we determined the factors that influence the rate of success of prism glasses. In a previous study, we had determined the efficacy of prism glasses in correcting PAET with respect to the maintenance of fusion and improvement of sensory status after one year of using prism glasses;[14] however, the long-term effect of prism glasses in reducing the angle of esodeviation has not been evaluated. Thus, in this study, we obtained the long-term results (greater than three years) of prismatic correction in patients with PAET.

In this study, 30.6% of PAET patients showed a residual esotropia of $\leq 10$ PD after 3 years of prism-wear and were defined as the Success group. Despite the fact that only 7.3% of patients weaned off prisms, a longer follow-up duration may increase the number of cases that do not require prisms in the Success group regarding the slow but continuous rate of decrease in esodeviation over time (Fig 1). Age of onset, age of initial visit, the time from onset to treatment, amblyopia, stereoacuity, or early fusional response at distance measured by the W4D test did not affect motor outcomes.

**Table 2. Comparison of characteristics between patients who weaned off prism glasses or not in the Success group.**

| | Prism-Weaned Group | Prism-Wearing Group | *P*-value |
|---|---|---|---|
| Number of patients | 9 (23.7%) | 29 (76.3%) | |
| Age at onset | 4.0±2.5 (1.2~9.9) | 3.5±2.0 (1.0~10.0) | 0.539[a] |
| $DCC_{Initial}$ (PD) | 10.3±3.4 (4~15) | 12.9±3.4 (6~20) | 0.059[a] |
| $NCC_{Initial}$ (PD) | 9.8±4.1 (1~15) | 13.7±6.5 (2~26) | **0.047[a]** |
| $DCC_{PAT}$ (PD) | 15.6±6.5 (10~28) | 17.7±4.7 (10~28) | 0.289[a] |
| $NCC_{PAT}$ (PD) | 10.4±4.0 (1~15) | 15.7±6.2 (3~28) | **0.007[a]** |
| Amblyopia | 3 (33.3%) | 11 (37.9%) | 0.565[b] |
| $SEQ_{Initial}$ (D) | +5.03±0.98 (+2.00~+7.00) | +4.02±1.40 (+1.50~+7.00) | 0.052[c] |
| $SEQ_{Final}$ (D) | +4.51±1.10 (+1.00~+7.00) | +3.57±1.74 (+0.56~+7.00) | 0.054[c] |
| Stereopsis (initial) | | | |
| Randot stereoacuity* (log arcsec) | 3.54±0.00 (3.54) | 2.19±0.17 (1.78~3.54) | 0.147[d] |
| $W4D_{PAT}$ normal at distance† | 4 (57.1%) | 15 (65.2%) | 0.515[b] |
| $W4D_{PAT}$ normal at near‡ | 5 (71.4%) | 13 (61.9%) | 0.509[b] |

*P*-values marked in bold indicate numbers that are significant on the 95% confidence limit; PD = prism diopters; D = diopters; PAT = prism adaptation test; $DCC_{Initial}$ = distant esodeviation with full hypermetropic correction before prism-wear; $NCC_{Initial}$ = near esodeviation with full hypermetropic correction before prism-wear; $DCC_{PAT}$ = the maximum angle of esodeviation at distance during the first 3 months of prism-wear with full hypermetropic correction; $NCC_{PAT}$ = the maximum angle of esodeviation at near during the first 3 months of prism-wear with full hypermetropic correction; $SEQ_{Initial}$ = spherical equivalent of cycloplegic refractive error on initial examination; $SEQ_{Final}$ = spherical equivalent of cycloplegic refractive error after 3 years of prism-wear; $W4D_{PAT}$ = Worth-4-dot test results during the first 3 months of prism-wear with full hypermetropic correction;

[a] Mann-Whitney U test;

[b] Fisher's exact test;

[c] Independent student t-test;

[d] Analysis of covariance (ANCOVA) test;

*N = 13 (3/9 of the prism-weaned group and 10/29 of prism-wearing group were tested due to young age);

†N = 30 (7/9 of the prism-weaned group and 23/29 of prism-wearing group were tested);

‡N = 28 (7/9 of the prism-weaned group and 21/29 of prism-wearing group were tested)

In our previous study on the efficacy of prism glasses in PAET, one-year success rates increased in patients without amblyopia and with good stereoacuity or good fusional response. [14] A few patients who were considered to have achieved success in the previous study were converted to the Failure group in the present study. This is mainly due to the difference in the definition of success, which was more strictly confined to a residual esotropia (the summation of the measured angle of esotropia wearing prism glasses and the amount of base-out prisms on both eyes) of ≤ 10 PD at distance in the present study. Moreover, given that a large portion of patients was unable to perform the stereoacuity tests due to their young age and poor cooperation, and also that these patients would likely have poor stereopsis, stereoacuity may have been overestimated by only including fully cooperative patients with good stereoacuity, particularly in the Failure group. In fact, 9 out of 11 patients without a measurement of stereoacuity at the initial exam finally showed nil stereoacuity in the Failure group.

Regarding the sensory response, Randot stereoacuity tested at near improved in 37.9% of patients (60.5% of the Success group and in 27.9% of the Failure group) at 3 years after using prism glasses. The sensory response became significantly better in the Success group compared

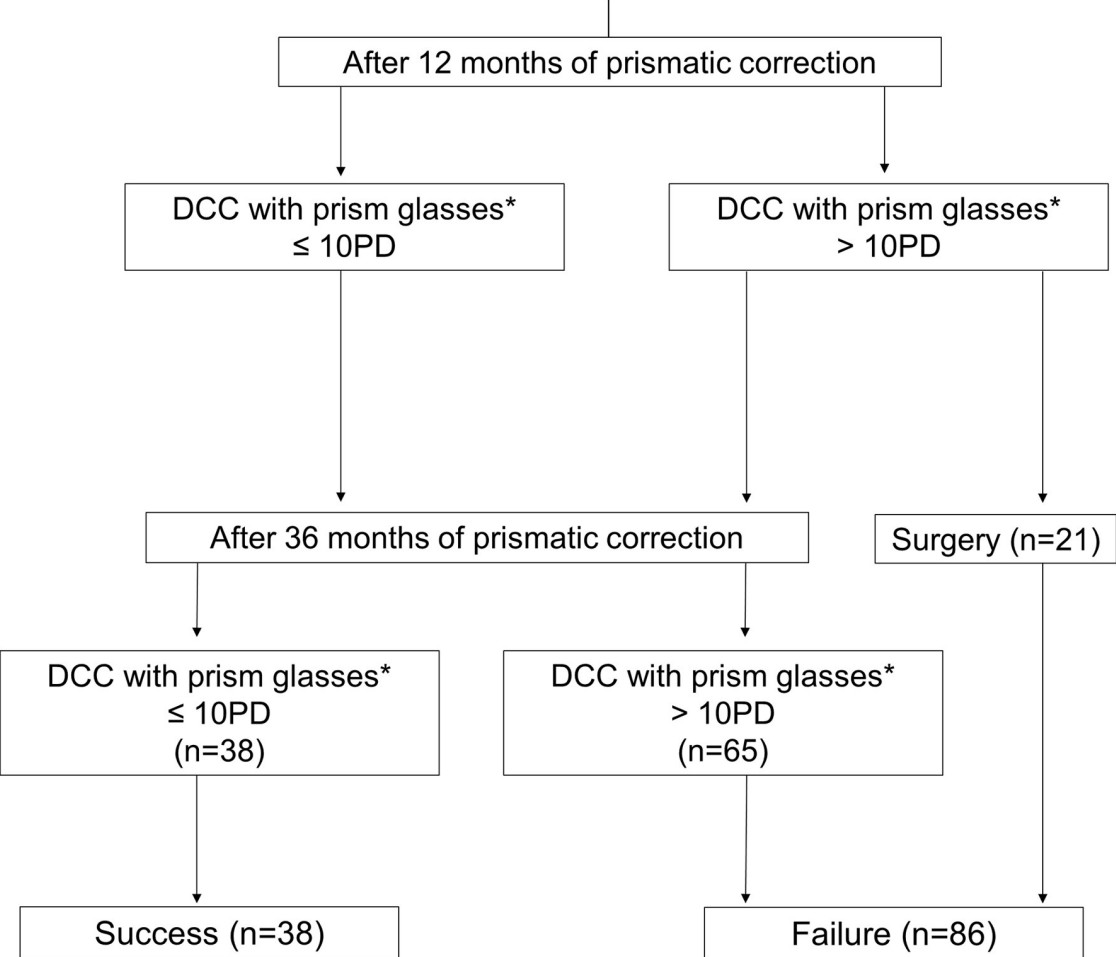

**Fig 3. Flow chart of results according to the response to prismatic correction.** *DCC with prism glasses = angle of esotropia measured by the simultaneous prism and cover test during distance viewing + amount of base-out prisms on both eyes.

to the Failure group only after 3 years, which was slower than the improvement in motor response that was apparent after 6 months.

In general, recovering high degree of stereoacuity is considered to be limited in esotropia, [19, 20] but many studies have shown improvement of stereopsis and sought factors affecting postoperative outcomes of stereoacuity. [21–24] Previous studies of congenital or infantile esotropia have shown that it is possible to obtain sensory and motor fusion by surgical alignment and prism glasses,[20, 21] and other studies on esotropia of adults also showed improvement in binocular function when the motor alignment was adjusted through surgery.[25]

In our previous study, PAET patients who maintained fusion by using prism glasses alone showed an improvement in stereoacuity after one year.[14] In infantile esotropia, shorter duration of misalignment and younger age at surgery affect good sensory response after surgery.[21] Iordanous et al.[24] showed that patients who maintained orthotropia or esotropia by less than 2 PD without consecutive exotropia after surgery showed better sensory prognosis in patients with PAET. Therefore, early achievement of fine motor alignment is considered to be necessary for the recovery of stereoacuity in esotropia. In this study, we showed that maintenance of fusion by using prism glasses had improved stereoacuity without surgical intervention. The rate of improvement was higher in the Success group than in the Failure group.

The amount of esodeviation started to show a significant difference between the two groups after 6 months. During the first 3 months of prism-wear, there was an increase in the angle of esodeviation in some cases, suggesting latent strabismus which is compatible with the increase of esodeviation frequently found after prism adaptation.[13] Therefore, the maximal angle of deviation during this period was determined as the baseline angle of esodeviation during prism-wear. The response to prism adaptation, which is the maximum amount of increase in esodeviation during this period, can be considered as a good predictor for long-term motor success of prism glasses. In patients who achieved motor success, 68% of patients showed less than 6 PD of increase in the distant angle of esodeviation after 3 months of prism adaptation.

One of the notable findings in this study is that only 9 patients (7.3%) finally weaned off prism glasses with only hypermetropic correction after 3 years. Patients who had a smaller angle of esodeviation after full hypermetropic correction at near before wearing prisms, and a smaller angle of latent esodeviation at near after prism adaptation were likely to take off prism glasses. Thus, prism glasses can be considered as an option for non-surgical treatment in some patients, though the proportion of patients that completely weaned-off prisms was small.

This study has some limitations. We defined the Success group as a residual angle of esotropia (the summation of the measured angle of esotropia wearing prism glasses and the amount of base-out prisms on both eyes) of ≤ 10PD with full hypermetropic correction and maintenance of fusion with prism glasses. Therefore, "success" does not mean complete weaning of prism glasses. Secondly, AC/A ratio were not measured in all patients. Hence, a subgroup analysis could not be performed in the heterogeneous patients with variable kinds of distance-near disparities. Additionally, we performed cycloplegic refraction using 1% cyclopentolate, but there has been a report that atropine refraction may have to be performed to detect latent strabismus in Asians who have darker iris.[26]

In conclusion, prism glasses for partially accommodative esotropia can be a non-surgical treatment option in selected patients who have a small angle of latent esodeviation revealed by prism adaptation and good sensory fusion at near with prism glasses. However, early surgery may also be advisable regarding the fact that weaning of prisms was achieved in only 7% of patients after 3 years.

## Supporting information

**S1 File. Minimal data set.** Clinical characteristics of the 124 children with partially accommodative esotropia with a residual esotropia of ≤ 20 prism diopters (PD) after full hypermetropic correction. All children were fitted with prism glasses for more than 1 year and followed-up for 3 years or more unless they underwent strabismus sugery.
(PDF)

 

## Author Contributions

**Conceptualization:** Jeong-Min Hwang.

**Data curation:** Hye Rim Choe, Hee Kyung Yang.

**Formal analysis:** Hye Rim Choe, Hee Kyung Yang, Jeong-Min Hwang.

**Investigation:** Hee Kyung Yang, Jeong-Min Hwang.

**Methodology:** Hye Rim Choe, Hee Kyung Yang, Jeong-Min Hwang.

**Project administration:** Jeong-Min Hwang.

**Resources:** Jeong-Min Hwang.

**Supervision:** Hee Kyung Yang, Jeong-Min Hwang.

**Validation:** Hee Kyung Yang, Jeong-Min Hwang.

**Visualization:** Hye Rim Choe, Hee Kyung Yang.

**Writing – original draft:** Hye Rim Choe, Hee Kyung Yang.

**Writing – review & editing:** Hee Kyung Yang, Jeong-Min Hwang.

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
