## [Decision Letter · Decision Letter 0]

29 Aug 2019

PONE-D-19-21942

Long-Term Outcomes of Prismatic Correction in Partially Accommodative Esotropia

PLOS ONE

Dear Dr. Hwang,

Thank you for submitting your manuscript to PLOS ONE. After careful consideration, we feel that it has merit but does not fully meet PLOS ONE’s publication criteria as it currently stands. Therefore, we invite you to submit a revised version of the manuscript that addresses the points raised during the review process.

We would appreciate receiving your revised manuscript by Oct 13 2019 11:59PM. To enhance the reproducibility of your results, we recommend that if applicable you deposit your laboratory protocols in protocols.io, where a protocol can be assigned its own identifier (DOI) such that it can be cited independently in the future. For instructions see: http://journals.plos.org/plosone/s/submission-guidelines#loc-laboratory-protocols

We look forward to receiving your revised manuscript.

Kind regards,

Ahmed Awadein, MD, Ph.D, FRCS

Academic Editor

PLOS ONE

Journal Requirements:

 [No].

Please provide an amended Funding Statement that declares *all* the funding or sources of support received during this specific study (whether external or internal to your organization) as detailed online in our guide for authors at http://journals.plos.org/plosone/s/submit-now.  

Please state what role the funders took in the study.  If any authors received a salary from any of your funders, please state which authors and which funder. If the funders had no role, please state: "The funders had no role in study design, data collection and analysis, decision to publish, or preparation of the manuscript."

[No].

Additional Editor Comments (if provided):

Reviewers' comments:

Reviewer's Responses to Questions

**Comments to the Author**

1. Is the manuscript technically sound, and do the data support the conclusions?

Reviewer #1: Yes

Reviewer #2: Partly

2. Has the statistical analysis been performed appropriately and rigorously? 

Reviewer #1: Yes

Reviewer #2: I Don't Know

3. Have the authors made all data underlying the findings in their manuscript fully available?

Reviewer #1: Yes

Reviewer #2: Yes

4. Is the manuscript presented in an intelligible fashion and written in standard English?

Reviewer #1: Yes

Reviewer #2: Yes

5. Review Comments to the Author

Reviewer #1: The authors presented a well-written manuscript entitled "Long-Term Outcomes of Prismatic Correction in Partially Accommodative Esotropia".

My comments to the authors:

1- the study was a retrospective study performed on children less than 10 years with partially accommodative esoropia seen from 2003-2012. The total number of children was 124, which is relatively small in a nine-year duration.

2- In the methodology section, the authors didnt mention measuring the AC/A ratio to any child.

3-In the results section the authors mentioned "Of the 86 patients in the Failure group, 21 patients (24.4%) received

surgery. What were the other alternatives for surgery for the failure group then?

Reviewer #2: Patient and methods: Prescription of prism glasses is not clear, please rewrite it with more simplicity.

Amblyopia cases must be discussed in more details (Degrees of amblyopia, duration of occlusion,…..)

Please mention the meaning of all abbreviations throughout the manuscript

(DCCPAT NCCPAT),(DCCinitial and NCCinitial)

6. PLOS authors have the option to publish the peer review history of their article (what does this mean?). If published, this will include your full peer review and any attached files.

Reviewer #1: No

Reviewer #2: No

---

## [Author Response · Author response to Decision Letter 0]

3 Nov 2019

Reviewer #1: The authors presented a well-written manuscript entitled "Long-Term Outcomes of Prismatic Correction in Partially Accommodative Esotropia".

My comments to the authors:

1- the study was a retrospective study performed on children less than 10 years with partially accommodative esotropia seen from 2003-2012. The total number of children was 124, which is relatively small in a nine-year duration.

→ Thank you for your comment. The majority of partially accommodative esotropia patients received operation within one year of follow-up. In South Korea, the treatment strategy is also influenced by the insurance system, as strabismus operation is covered by the national insurance only for patients under 10 years old, and many parents want to have surgery at a younger age given the possibility of reoperation. Therefore, the number of patients satisfying our inclusion criteria, “who were fitted with prism glasses for 3 years or more to correct their residual esodeviation”, is relatively small considering the long study duration. 

2- In the methodology section, the authors didnt mention measuring the AC/A ratio to any child.

→ We measured the AC/A ratio at the initial examination by the clinical method evaluating distance-near relationship. High AC/A ratio was defined when the near esodeviation was ≥ 10PD larger than the distant esodeviation. A total of 17 patients (13.7%) showed a high AC/A ratio at the initial examination. We added this in the methods and results section.

3-In the results section the authors mentioned "Of the 86 patients in the Failure group, 21 patients (24.4%) received surgery. What were the other alternatives for surgery for the failure group then?

→ While 21 patients received surgery after more than 1 year of wearing prism glasses, the other patients who did not meet the surgical criteria maintained their glasses with prisms ranging from 12 PD to 30 PD (including fresnel prisms). These patients showed a relatively stable angle of esodeviation with prism glasses, because patients who presented with a rapid increase of esodeviation received surgery. On the other hand, among the success group who were still wearing small angles of prisms, patients who wanted to take off prism glasses selected to receive surgery after more than 3 years of prism wear. We added this in the methods section.

We deeply appreciate your precious time and effort for the review.

 

Reviewer #2: Patient and methods: Prescription of prism glasses is not clear, please rewrite it with more simplicity.

→ Prism gasses were initially prescribed according to the residual angle of esodeviation during distant fixation (DCCinitial) and cycloplegic refractive errors. The amount of prisms were changed whenever the angle of esotropia was changed.

For example, if a patient’s cycloplegic refractive errors were +3.00 Dsph in both eyes, and the esodeviation measured with glasses (+3.00 Dsph OU) was 12 prism diopters during distant fixation, then prism glasses of +3.00 Dsph with 6 base-out prisms were prescribed for each eye. At the next follow up, if the patient revealed an esotropia of 4 prism diopters wearing those glasses with the simultaneous cover test, the amount of prisms were increased to 8 base-out prisms for each eye. We clarified this in the methods section. 

Amblyopia cases must be discussed in more details (Degrees of amblyopia, duration of occlusion,…..)

→ At the initial exam, 27 patients (21.8%) had amblyopia and full-time occlusion except one day in a week was prescribed in 21 patients (16.9%) initially. For 6 patients (4.8%) who were under 4 years old, part time occlusion therapy was prescribed for 2 to 6 hours a day. Occlusion was tapered according to their visual response, and 8 of patients (6.5%) continued part time occlusion therapy at the last follow-up. We added this in the results section.

Please mention the meaning of all abbreviations throughout the manuscript

(DCCPAT and NCCPAT),(DCCinitial and NCCinitial)

→ For the residual angle of esodeviation after full hypermetropic correction at initial exam, we used abbreviations “DCCinitial and NCCinitial”. The first prism glasses were prescribed according to “DCCinitial”. 

→ The maximum angle of baseline esodeviation in the first 3 months after prism wear was defined as “DCCPAT or NCCPAT”, which indicate the stabilized angles of esodeviation by prism adaptation. These values were usually greater than “DCCinitial and NCCinitial” due to manifestation of latent strabismus. We used these values as baseline angles for evaluating the effect of prism wear.

We deeply appreciate your precious time and effort for the review.

---

## [Editor Report · Decision Letter 1]

11 Nov 2019

Long-Term Outcomes of Prismatic Correction in Partially Accommodative Esotropia

PONE-D-19-21942R1

Dear Dr. Hwang,

We are pleased to inform you that your manuscript has been judged scientifically suitable for publication and will be formally accepted for publication once it complies with all outstanding technical requirements.

With kind regards,

Ahmed Awadein, MD, Ph.D, FRCS

Academic Editor

PLOS ONE
---

## [Editor Report · Acceptance letter]

20 Nov 2019

PONE-D-19-21942R1 

Long-Term Outcomes of Prismatic Correction in Partially Accommodative Esotropia 

Dear Dr. Hwang:

I am pleased to inform you that your manuscript has been deemed suitable for publication in PLOS ONE. Congratulations! Your manuscript is now with our production department. 

With kind regards,

on behalf of

Dr. Ahmed Awadein 

Academic Editor

PLOS ONE